# Assessment of the Degradation Potential and Genomic Insights towards Phenanthrene by *Dietzia psychralcaliphila* JI1D

**DOI:** 10.3390/microorganisms9061327

**Published:** 2021-06-19

**Authors:** Janardhan Ausuri, Giovanni Andrea Vitale, Daniela Coppola, Fortunato Palma Esposito, Carmine Buonocore, Donatella de Pascale

**Affiliations:** 1Institute of Biochemistry and Cell Biology (IBBC), National Research Council, Via Pietro Castellino 111, 80131 Naples, Italy; janardhan.ausuri@ibbc.cnr.it (J.A.); carmine.buonocore@szn.it (C.B.); 2Department of Marine Biotechnology, Stazione Zoologica Anton Dohrn, Villa Comunale, 80121 Napoli, Italy; giovanniandrea.vitale@szn.it (G.A.V.); daniela.coppola@szn.it (D.C.); fortunato.palmaesposito@szn.it (F.P.E.)

**Keywords:** *Dietzia*, bioremediation, polycyclic aromatic hydrocarbons (PAHs), whole-genome sequencing

## Abstract

Extreme marine environments are potential sources of novel microbial isolations with dynamic metabolic activity. *Dietzia psychralcaliphila* J1ID was isolated from sediments originated from Deception Island, Antarctica, grown over phenanthrene. This strain was also assessed for its emulsifying activity. In liquid media, *Dietzia psychralcaliphila* J1ID showed 84.66% degradation of phenanthrene examined with HPLC-PDA. The identification of metabolites by GC-MS combined with its whole genome analysis provided the pathway involved in the degradation process. Whole genome sequencing indicated a genome size of 4,216,480 bp with 3961 annotated genes. The presence of a wide range of monooxygenase and dioxygenase, as well as dehydrogenase catabolic genes provided the genomic basis for the biodegradation. The strain possesses the genetic compartments for a wide range of toxic aromatic compounds, which includes the *ben*ABCD and *cat*ABC clusters. COG2146, COG4638, and COG0654 through COG analysis confirmed the genes involved in the oxygenation reaction of the hydrocarbons by the strain. Insights into assessing the depletion of phenanthrene throughout the incubation process and the genetic components involved were obtained. This study indicates the degradation potential of the strain, which can also be further expanded to other model polyaromatic hydrocarbons.

## 1. Introduction

The increasing chemical and biological pressure induced by human activities on marine ecosystems has caused adverse effects on marine health. Over the last few decades, there has been increased global awareness towards remediating and renovating such toxic marine environments. Among others, polycyclic aromatic hydrocarbons (PAHs) is of major concern due to their toxic, genotoxic, mutagenic properties [1]. Polyaromatic hydrocarbons are composed of aromatic rings arranged in a linear, angular, or cluster manner. Depending on the aromatic rings, PAHs can range from two benzene rings (naphthalene) to seven benzene rings (coronene). These hydrocarbons possess various physiochemical properties, such as electrochemical stability, persistency, structural angularity, and hydrophobicity, which make them ideal candidates as intermediaries in thermosetting plastics and lubricating materials [2]. Concerning the toxic nature of PAHs, 16 of these substances have been enlisted as priority pollutants by the United States Environmental Protection Agency (US EPA) [3]. The reclamation and remediation of such polluted sites is of high priority. Many physio-chemical techniques, such as UV-oxidation and solvent extraction, are in practice, but they come at a high cost and regulatory burden [4]. Microorganisms have developed numerous metabolic strategies in utilizing these hydrocarbons as their carbon source [5]. PAH degradation constitutes a catabolic funnel where the formation of dihydrodiols is achieved by dioxygenases, which are then converted to diol intermediates by dehydrogenases, subsequently yielding catechol-based intermediates by the action of intradiol/extradiol dioxygenases, ultimately leading to formation of the TCA cycle intermediates [6,7].

Of all the available PAHs, phenanthrene is considered as a prototype model for biodegradation studies as its structural backbone is found in other PAHs and often serve as an indicator in the detection of PAH contamination [6]. Phenanthrene is a Low Molecular Weight Polycyclic Aromatic Hydrocarbon (LMWPAH), and it consists of three benzene rings arranged in angular fashion consisting of a Bay region and K-region, which are considered to be highly reactive sites of the molecule [8]. Evans and collaborators (1965) first reported the bacterial degradation of phenanthrene by *Pseudomonas aeruginosa*, following a ring-fission manner [9]. Many bacteria belonging to genera *Arthobacter*, *Stenotrophomonas*, *Acidovorax*, *Rhodococcus*, *Brevibacterium*, and *Burkholderia* have been reported for phenanthrene degradation [10,11]. Whole genome sequencing allows a comprehensive methodology into the genomic features of bacteria. The genome annotation gives a complete understanding of the dedicated aromatic degrading genes coding for the enzyme, dioxygenase, monooxygenase, dehydrogenase, and decarboxylases [12,13]. Various catabolic pathways have been described for phenanthrene degradation [12,14], where the parent hydrocarbon is shredded into its corresponding catechol-derived daughter products. This combined approach is a powerful tool in identifying suitable candidates to construct a bacterial consortium for in situ bioremediation techniques [15]. Apart from mentioning the involved genes obtained from genome sequencing, the analysis of Clusters of Ortholog Groups (COGs), conserved domains (CDs), and Pfams are of growing interest, helping in unravelling the dynamic genetic components present in bacteria to survive in extreme conditions [16].

The genus of *Dietzia* is reported for its mineralization capacities of aliphatic hydrocarbons, such as n-alkanes ranging from n-C_6_–n-C_26_ [17] to n-C_12_–n-C_38_ alkanes [18] due to the presence of the *alkB* gene in their genetic system [19]. Very few studies have been reported for PAH degradation for the genus *Dietzia*, but which include Al-Awadhi et al. [20] for phenanthrene degradation and Gurav et al. [21] for naphthalene degradation in a consortium. Herein, we report the isolation and biochemical screening of a phenanthrene-degrading bacterium. Afterwards, a quantitative assessment of the phenanthrene degradation was studied. Whole genome sequencing revealed that the selected bacterium might belong to a novel strain belonging to the *Dietzia psychralcaliphila* species, and, finally, the metabolic pathways of the phenanthrene degradation were described.

## 2. Materials and Methods

### 2.1. Sediment Collection

Marine sediments were collected from various locations on, or surrounding, Base Antarctica Española Gabriel de Castilla, Deception Island, Antarctica. The samples were collected in sterile 50 mL falcon tubes from a shallow depth between 3 and 14.5 m, 62°58′34.6″ S 60°40′31.7″ W, at temperatures between 2 and 4 °C. The samples were transported and stored at −20 °C for future usage.

### 2.2. Media and Chemicals

Mineral Salt Medium (MSM) (per liter): K_2_HPO_4_ 1.8 g; NH_4_Cl 2.5 g; MgSO_4_·7H_2_O 0.2 g; NaCl 0.1 g; FeSO_4_·7H_2_O 0.01 g; yeast extract 0.5 g; trace metal solution 1 mL; the MSM liquid media was adjusted to pH 6.9. For solid media, 15 g of bacteriological agar were added. The trace metal composition (per liter) consisted of ZnSO_4_ 0.29 g; CaCl_2_ 0.24 g; CuSO_4_ 0.25 g; MnSO_4_ 0.17 g; it was filtered with a 0.22 µm pore membrane. Luria-Bertani (LB) broth was composed (per liter) of tryptone 10.0 g; yeast extract 5.0 g; NaCl 5.0 g. Phenanthrene was obtained from Sigma-Aldrich (Darmstadt, Germany) with a purity of 98%. Stock solution of phenanthrene was 4 g dissolved in 1 L diethyl ether.

### 2.3. Isolation of Phenanthrene Degrading Bacteria

One gram of soil with 1 mL of artificial sea water was centrifuged at 2000 rpm for 30 s. The supernatant was collected and serially diluted ten folds until 10^−6^ dilutions. The dilutions were made to spread over MSM agar plates containing 1.0 mM of phenanthrene from the stock solution prepared. The agar plates with phenanthrene were incubated at 20 °C for 3 weeks to observe any microbial growth. The isolate forming a clear zone (indicating phenanthrene utilization) was picked. The isolate was streaked with different concentrations of phenanthrene (2.0, 3.0, 4.0, and 5.0 mM) on MSM agar plates in triplicates and incubated at 20 °C. Based on growth over plates, the isolate was picked and transferred to liquid media in the presence of phenanthrene. The isolate was re-streaked in LB agar plates and grown in liquid LB for further studies. The isolated strain was cryopreserved at −80 °C with 20% (*v*/*v*) glycerol.

### 2.4. Molecular and Phylogenetic Analysis

Bacterial genomic DNA for the isolates was extracted using the “GenElute Bacterial genomic DNA kit” (Sigma-Aldrich, Darmstadt, Germany). A 2 mL culture broth from each isolate growing at the exponential phase was taken and the DNA was extracted by following the manufacturer’s protocol. From the extracted genomic DNA, amplification via PCR of the 16S rRNA sequence was performed. The 16S rRNA genes were amplified using primers 27F (5′-AGAGTTTGATCMTGGCTCAG-3′) and 1492R (5′-GGTTACCTTGTTACGACTT-3′). PCR was carried out in a total volume of 50 µL containing 1× reaction buffer, 150 µM MgCl_2_, 250 µM of each deoxy nucleoside triphosphate, 2.0 U of PolyTaq DNA polymerase, and 0.5 µM primers (Thermo Fisher Scientific, Darmstadt, Germany). The concentration and purity of the extracted DNA was verified from the absorbance values as detected by Nanodrop UV spectrophotometer 2000 c (Thermofisher Scientific). Amplicons were purified with the GeneAll kit (Thermofisher Scientific, Darmstadt, Germany) and sequenced by Eurofins Genomics (Ebersberg, Germany). The phylogenetic relationship was established by comparing 16S rRNA sequence with the GeneBank database, National Centre for Biotechnology Information (NCBI). The evolutionary relationship of taxa was conducted by MEGA X [22]. The bootstrap method was used to construct the phylogenetic tree.

### 2.5. E_24_ Assay

To estimate the production of the biosurfactants, the E_24_ assay was conducted. Two mL of various hydrocarbons like benzene, xylene, toluene, diesel, and cooking oil were added to 2 mL of supernatant of bacterial culture. The suspension was vortexed at high speed for 2 min vigorously and left 24 h undisturbed at room temperature. Tween 20^®^ of 0.5% *v*/*v* was used as positive control. The emulsification index (E_24_) was calculated as
(1)$$\mathrm{Emulsification}\;\mathrm{index}=\frac{\mathrm{Height}\;\mathrm{of}\;\mathrm{surfactant}\;\mathrm{layer}\;(\mathrm{mm})}{\mathrm{Total}\;\mathrm{height}\;\mathrm{of}\;\mathrm{liquid}\;(\mathrm{mm})}\;\times \;100$$

### 2.6. Biodegradation Analysis of Phenanthrene

#### 2.6.1. Quantification Analysis by High-Performance Liquid Chromatography (HPLC)

The bacterial cultures that had reached the late exponential phase, grown at 25 °C and 180 rpm, were centrifuged at 6000 rpm for 10 min, washed twice, and re-suspended in MSM media to prevent the interference of any carbon source that formed the cell suspension. In a 250 mL Erlenmeyer flask, 2.0 mM of phenanthrene was dissolved in diethyl ether. After evaporation of ethereal solution, MSM media was added and autoclaved at 120 °C for 20 min. The media was supplemented with 1% (*v*/*v*) of the cell suspension, OD_600_ = 1.0, with a total volume of 100 mL. The flasks without cell suspensions formed the abiotic control. Conditions were ensured that the inoculum had enough space for exchange of oxygen to that of the surroundings. The flasks were incubated at 180 rev min^−1^ at 25 °C. To assess the residual phenanthrene, 5 mL of culture was withdrawn at different time intervals and extracted with twice the volume of n-hexane, after which they were dried with a rotary evaporator (R-100, BUCHI, Flawil, Switzerland). The obtained extracts were run on high-performance liquid chromatography (HPLC) (Agilent Technologies 1100 series) at a 254 nm wavelength based on signals of photodiode array detector (PDA). Samples of 10 µL were injected into a reverse phase C 18 LC column and run in an isocratic gradient for 18 min (Buffer A: water + 0.1% trifluoracetic acid (TFA); Buffer B: acetonitrile + 0.1% TFA) with a flow rate of 1 mL/min. The standard concentration of phenanthrene was prepared in n-hexane. To calculate the percentage of degradation of phenanthrene, the following formula was used.
(2)$$\%\mathrm{Degradation}\;\mathrm{of}\;\mathrm{phenanthrene}=\frac{C_{0}{-C}_{t}}{C_{0}}\;\times \;100$$
where C_o_ is the initial concentration and C_t_ is the final concentration of phenanthrene.

The biodegradation rate (Rx) of phenanthrene is the concentration of phenanthrene utilized divided by the time, expressed as (mM/day), and was calculated by the following formula [23].
(3)$$R_{x}=-\frac{{(N}_{0}{-N}_{t})}{{(T}_{0}{-T}_{t})}$$
where N_0_ and N_t_ are the initial and final concentrations of phenanthrene, respectively, and T_0_ and T_t_ are the initial and final days of inoculation, respectively.

#### 2.6.2. Pathway Analysis by Gas Chromatography–Mass Spectrometry (GC-MS)

To examine the pathway followed by phenanthrene, the culture was constructed as described in Section 2.6.1. At the end of 7 days, the culture was extracted twice by adding 50 mL of ethyl acetate and shaking vigorously for the separation of phases. The organic phase was chosen and dried by rotary evaporator (R-100, BUCHI, Flawil, Switzerland). The dried extracts were suspended in 1 mL methanol (HPLC grade) for GC-MS analysis. The metabolites were identified by Thermo scientific DSQ II series single quadrupole GC-MS. The GC-MS was operated in positive mode with an ionization energy 70 eV. The ion source temperature was 250 °C. Mass units were monitored in the range between 35 and 1000 m/z. The injection volume adopted was 0.7 µL.

### 2.7. Whole-Genome Sequencing

#### 2.7.1. Whole Genome Sequencing and De Novo Assembly

The genomic DNA of the selected strain was extracted using the “GenElute Bacterial genomic DNA kit”. The genomic DNA of >1 µg was used for assembly purposes. The DNA had been sent to Macrogen Europe B.V., The Netherlands. The sequence library was prepared by random fragmentation of the DNA samples. Illumina SBS technology was used for genome sequencing that resulted in highly accurate base-by-base sequencing. The assembled genome was validated using mapping strategy and BUSCO analysis.

#### 2.7.2. Genome Annotation

The assembled genome was visualized circularly using the online tool CGviewer (toolhttps://beta.proksee.ca/projects (accessed on 30 May 2020)). The assembled genome was annotated using PROKKA annotation v1.14.6 as described by [24] and with Rapid Annotation using the Subsystem Technology (RAST) server [25]. Functional genome annotation was performed with “emapper version: emapper-1.0.3-35-g63c274b emapper DB: 2.0” [26,27]. The homologous unique genes in the whole genome were analyzed by BLAST search on the NCBI database. An integrated interface for computational identification and visualization of genomic islands (GIs) was done by the Island Viewer 4 webserver (http://www.pathogenomics.sfu.ca/islandviewer/ (accessed on 24 September 2020)) [28]. IslandPick, IslandPath-DIMOB, SIGI-HMM, and Islander were used by Island Viewer 4 for GIs prediction. The genome of the strain under study was compared with the *D. psychralcaliphila* strain ILA-1. The pathway analysis, ortholog assignment, and mapping of genes were done by KEGG automatic annotation server (KAAS) [29]. All the gene sequences were analyzed against the KEGG database using BLASTP. Furthermore, to assess the presence of various domains from the protein-sequences annotated, for NCBI’s Conserved Domain Database (CDD), SMART (Simple Modular Architecture Research Tool), PRK (Protein Clusters version 6.0 from NCBI CDD v3.16), TIGRFAMs, NCBIfam, and Kbase platform were used with the domains application (https://narrative.kbase.us/#catalog/apps/DomainAnnotation/annotate_domains_in_a_genome/46df12c6f8b16ae25df386c05e78c94292121db2 (accessed on 9 May 2020)). The whole genome project was submitted under biosample accession SAMN17112250 and the BioprojectID was assigned as PRJNA686264 in the NCBI database.

#### 2.7.3. Phylogenetic Relationship

After the genome assembly, relative species or evolutionarily different species relating to the strain was analyzed. To provide more accuracy, the species core genome (SCG) was analyzed among 35 genomes belonging to *Dietzia*. After which, the potential genomic novelty was investigated using the publicly available FastANI tool, as prescribed by [30].

## 3. Results

### 3.1. Isolation, Molecular Identification, and Phylogenetic Relationship of Phenanthrene-Degrading Bacteria

To demonstrate the ability of bacterial growth over phenanthrene, soil samples collected from Deception Island, Antarctica, were tested in MSM agar plates with 1.0 mM of phenanthrene. These were compared with a non-supplemented phenanthrene control. The focus leaned towards isolates that could grow only in laboratory conditions. After 3 weeks of incubation at 20 °C, it was interesting to see that only one colony grew on the original dilution and none on the increased dilution plates. The phenanthrene utilization was confirmed by the presence of a clear zone around the colony. Moreover, the growth of the isolate was evaluated on different concentrations of phenanthrene (2.0, 3.0, 4.0, and 5.0 mM). On the plates of different concentration of phenanthrene, based on growth visualized after 7 days, 2.0 mM was chosen for further experiments as the other plates failed to show any growth. Through 16S rRNA sequencing, the isolate was identified closely related with *D. psychralcaliphila*. The evolutionary history of *D. psychralcaliphila* JI1D was inferred using the Neighbor-Joining method [31]. The bootstrap consensus tree inferred from 1000 replicates was taken to represent the evolutionary history of the taxa analyzed [32]. The analysis is depicted in Figure 1. The strain JI1D matched 97% to *D. psychralcaliphila.* When investigated for potential genomic novelty among the genus *Dietzia* by using the FastANI tool, *D. psychralcaliphila* JI1D in this study might represent a novel strain of the genome GCA_003096095.1 (97.93%) belonging to *D. psychralcaliphila* (Appendix A).

Branches corresponding to partitions reproduced in less than 50% bootstrap replicates are collapsed. The percentage of replicate trees in which the associated taxa clustered together in the bootstrap test (1000 replicates) are shown next to the branches. The evolutionary distances were computed using the Kimura 2-parameter method [33] and are in the units of the number of base substitutions per site. This analysis involved 30 nucleotide sequences. All ambiguous positions were removed for each sequence pair (pairwise deletion option). There were a total of 1531 positions in the final dataset.

### 3.2. E_24_ Assay

An E_24_ assay was performed to evaluate the biosurfactant production ability of the bacterium. The E_24_ assay when tested with a wide range of hydrocarbons showed maximum efficiency (ranging around 60%) using xylene, followed by benzene and toluene. Diesel and cooking oil emulsified above 40%. The E_24_ index of the positive control for xylene, toluene, and benzene was 65%, followed by diesel at 63% and that of the cooking oil was 56%. Biosurfactant production was indicated by the E_24_ assay (Figure 2).

### 3.3. Degradation and Identification of Phenanthrene Degradation Products

The biodegradation of phenanthrene by *D. psychralcaliphila* JI1D was quantified over time by HPLC, as described in Section 2.6.1. The residual concentration of phenanthrene in the abiotic control flasks on an average throughout the course of experiment was found to be 99.97 ± 0.00045%, indicating negligible or no loss of phenanthrene. At the end of 7 days of incubation, the culture was characterized with increased turbidity, emulsification of the phenanthrene crystals. The residual phenanthrene after 11 days was found to be 0.306 ± 0.014 mM, showing 84.66% of the phenanthrene was efficiently degraded (Figure 3). The biodegradation rate (R_x_) of phenanthrene was calculated to be 0.1545 mM/day.

The obtained dried extracts suspended in methanol was analyzed by GC-MS. The software Xcalibur was used to interpret the results. The peaks of the extracts were analyzed by comparing with NIST (National Institute of Standards and Technology) mass spectral library. The presence of phthalic acid at a retention time (RT) of 12.22 min and salicylic acid peak at 6.89 min suggests that *D. psychralcaliphila* JI1D degrades phenanthrene through both the o-phthalic acid pathway and naphthalene degradation pathway (and Figure 4). Initial metabolic intermediates involved in the upper catabolic pathway were not observed.

### 3.4. Whole Genome Sequencing and Genome Assembly

The extracted genomic DNA, of high quality > 1 µg, was sequenced using Illumina SBS technology. Prior to genome assembly, coverage, heterozygosity, and genome size were estimated using k-mer analysis. The de novo assembly was performed by various k-mer analyses using SPAdes. To assess the completeness of the genome assembly, a Benchmarking Universal Single-Copy Orthologs (BUSCO) analysis was performed based on evolutionarily informed expectations of the gene content from near-universal single-copy orthologs. The results are provided in Appendix A. Filtered reads were aligned against the assembled genome and their insert sizes were estimated for validation. The results are provided in Appendix A. This de novo assembly yielded a draft genome consisting of 52 contigs adding up to 4,216,480 base pairs (bp). In this, 601,190 bp was the longest while 1026 bp was the shortest contig, with the average length of each contig being 81,086 bp. The base contents of the contigs were obtained as listed in Appendix A. Upon assembly, validation was performed by mapping the results and the necessary statistics were calculated and presented in Table 1

### 3.5. Genome Annotation

The assembled genome was annotated, and its circular view was presented in Figure 5. BLAST1 represents the region of the reported *D. psychralcaliphila* ILA1 complete genome. The circular view displays the GC content, GC Skew+, GC Skew-, and RNAs. The genome annotation via PROKKA pipeline yielded 3902 CDSs. The strain possessed 55 tRNAs, 4 rRNAs, and 21 ncRNAs, as presented in Table 2.

When annotated through the RAST server, 107 genes were found involved in the metabolism of the aromatic compounds, out of which 69 were involved in the peripheral pathways for catabolism and 38 for metabolism of the central aromatic intermediates, as shown in Figure 6 and Appendix A.

The functional annotation of the whole genome was performed by eggNOG-Mapper by assigning a minimum hit e-value of 0.001, minimum hit bit-score of 60, and minimum % of query coverage of 20. The predominant classes of COGs belonged to unknown functions (16.89%), followed by lipid transport and metabolism (9.48%), transcription (8.74%), amino acid transport and metabolism (8.23%), and inorganic ion transport and metabolism (7.01%) when normalized with the total number of entries, as shown in Figure 7, with details in Appendix A. The domain annotation was performed through All-1.0.3 and 14,300 protein domains were annotated. Of which, CDD counted for 3480 domains, TIGRFAMs for 760 domains, PRK for 4855 domains, and SMART for 221 domains.

### 3.6. Genomic Islands

Genomic islands are gene clusters that are acquired through a horizontal gene transfer phenomenon. A total of 29 genomic islands were predicted in the genome of *D. psychralcaliphila* JI1D using Island Viewer 4 [28] (Figure 8). The total of genomic islands comprised 637,119 bp encoding 573 genes. The size range varied between 4 kbp and 165 kbp, encoding genes for transcription regulators, protein metabolism, transporter of sugar, and stress response. Interestingly, genes encoding proteins relative to aromatic compound degradation were also found. This shows that the strain is dynamic and can survive in extreme environments.

### 3.7. Proposed Pathway of Phenanthrene Degradation

The genome sequence of the strain *D. psychralcaliphila* JI1D was analyzed to obtain the catabolic genes that are supposed to be involved in phenanthrene degradation. Several genes were acquired using the KEGG automatic annotation server (KAAS) distributed across the genome displaying the capacity of the strain to utilize phenanthrene.

Within the annotated genes, 31 monooxygenases and 12 dioxygenases were present. To list a few key enzymes, the class of enzyme belonging to the oxygenase, aromatic ring hydroxylating type included p-cumate 2,3-dioxygenase system large oxygenase component (DOCFNAII-01294) and 2-halobenzoate 1,2dioxygenase alpha subunit (DOCFNAII-02907), and the aromatic ring cleaving type included catechol 1,2-dioxygenase (DOCFNAII-02908, DOCFNAII-00188) and protocatechuate 4,5-dioxygenase (DOCFNAII-02208). The monooxygenases belonging to Baeyer–Villiger monooxygenase (DOCFNAII-00841, DOCFNA-II00950, DOCFNAII-01790, DOCFNAII-03107), alkane 1-monooxygenase (DOXFNAII-01207), 2,5-diketocamphane 1,2-monooxygenase (DOCFNAII-01392), flavin-dependent monooxygenase (DOCFNAII02134), 3-ketosteroid-9-alpha-monooxygenase, ferredoxin reductase component (DOCFNAII02139), and alkanal monooxygenase alpha chain (DOCFNAII03104) were present. Apart from oxygenase, 1,2-dihydroxycyclohexa-3,5-diene-1-carboxylate dehydrogenase (DOCFNAII02904), 2-hydroxychromene-2-carboxylate isomerase/DsbA-like thioredoxin domain (DOCFNAII00917), 4-hydroxy-tetrahydrodipicolinate synthase (DOCFNAII-03745), 4-hydroxy-2-oxovalerate aldolase (DOCFNAII02141), 4,5-dihydroxyphthalate decarboxylase (DOCFNA11-02859), 3-(3-hydroxy-phenyl)propionate hydroxylase (salicylate) (DOCFNAII02881), and EPTC-inducible aldehyde dehydrogenase (DOCFNAII01122) were also present. Members of genes along the aromatic central metabolic pathway were identified, such as benzoate membrane transport protein *ben*E (DOCFNAII02900), MFS transporter *ben*K (DOCFNA1102903), dihydroxycyclohexadiene carboxylate dehydrogenase *ben*D (DOCFNAII02904), benzoate 1,2-dioxygenase reductase component *ben*C (DOCFNAII02905), benzoate 1,2-dioxygenase beta subunit *ben*B (DOCFNAII02906), muconate cycloisomerase *cat*B (DOCFNAII02909), muconolactone D-isomerase *cat*C (DOCFNAII02910), acetyl-CoA-transferase atoB (DOCFNAII02911), IclR family transcriptional regulator *pca*R (DOCFNAII02912), 3-oxoadipate enol-lactonase *pca*D (DOCFNAII02913), 3-oxoxacid CoA-transferase subunit A *sco*A (DOCFNAII02914), 3-oxoacid CoA-transferase subunit B *sco*B (DOCFNAII02915), and 4-carboxymuconolactone decarboxylase *pca*C (DOCFNAII0072). The presence of catabolic genes, such as peptidoglycan glycosyltransferase *Fts*W (DOCFNAII_01859), phosphomannomutase (DOCFNAII_02504, DOCFNAII_02532), glucose-1-phosphate thymidylyltransferase (DOCFNAII_03623, DOCFNAII_0173), dTDP-glucose 4,6-dehydratase (DOCFNAII_01735), dTDP-4-dehydrorhamnose 3,5-epimerase (DOCFNAII_01728), dTDP-4-dehydrorhamnose reductase (DOCFNAII_01733), malonyl CoA-acyl carrier protein transacylase (DOCFNAII_01771), glycosyl transferase (DOCFNAII_01686), and acyltransferase (DOCFNAII_01742, DOCFNAII_03350) displayed the ability of the strain in producing biosurfactants.

This is the first time that the approach of WGS has been reported to unravel the probable phenanthrene degradation pathway for the *D. psychralcaliphila* JI1D (Figure 9). Only the main enzymes involved in the degradation mechanisms, such as dioxygenase and dehydrogenases, have been mentioned. The presence of two aromatic ring hydroxylating dioxygenase alpha subunit genes (DOCFNAII_02907, DOCFNAII_01294) suggest they could be involved in the initial oxidation reaction from phenanthrene to its corresponding cis-arenedihydrodiols. The identified genes are also believed to be involved in consecutive aromatic ring hydroxylating steps. The family of 3-oxoacyl- (acyl carrier protein) reductase (DOCFNAII_00095, DOCFNAII_00124) consists of short-chain dehydrogenase reductases (SDRs). The presence of ring cleaving dioxygenases (DOCFNAII_00189, DOCFNAII_02908) could reveal the degradation of phenanthrene in the lower pathway of the catabolic funnel. The presence of catabolic genes 2-polyprenyl-6-methoxyphenol hydroxylase-like oxidoreductase (DOCFNAII_02881, DOCFNAII_03605) belonging to the salicylate hydroxylase (1.14.13.1) makes it evident that the strain also has the potential of mineralizing phenanthrene through the naphthalene degradation pathway. The presence of 4-hydroxy-tetrahydrodipicolinate synthase (DOCFNAII_00869, DOCFNAII_00143) could involve the transformation of trans-o-hydroxy benzylidene pyruvate to salicylaldehyde, ultimately leading to formation of salicylate. This is homologous in function with the o-phthalic acid pathway of phenanthrene degradation where cis-2′-carboxybenzyl pyruvate will be transformed to 2-carboxybenzaldehyde, leading to formation of phthalate. The gene 2-carboxybenzaldehyde dehydrogenase (*Phd*K) (DOCFNAII_03312) aids in formation of phthalate. Both pathways forming phthalate and salicylate lead to the benzoate degradation pathway where the intermediates are cleaved in either the ortho- or meta- way to enter the tricarboxylic acid cycle.

## 4. Discussion

Aquatic and terrestrial ecosystems are subjected to high anthropogenic pressure that leads to accumulation of toxic contaminants with potential detrimental consequences. Extreme marine environments are a rich source of diverse communities of microorganisms that can be employed in the cleaning-up process of contaminated sites [34]. Microorganisms display remarkable catabolic activities and expedite the remediation of such contaminated environments.

In this study, sediments originating from Deception Island, Antarctica, were investigated for bacteria growing under laboratory conditions utilizing the PAH phenanthrene. Earlier studies confirmed that bacterial isolates growing on PAH-substrates can be co-related to their assimilation of PAHs [35]. *D. psychralcaliphila* is a Gram-positive, non-motile, aerobic bacterium [36] showing growth on aliphatic hydrocarbons. The genus *Dietzia* has been reported to degrade a wide range of hydrocarbons, including aliphatic, aromatic, and diesel oil [8]. Several draft/complete genomes have been reported involving the hydrocarbon degradation [17,18,37] of the genus *Dietzia*.

Whole genome sequencing of *D. psychralcaliphila* JI1D was performed to understand the catabolic genes probably involved in PAH degradation. The interconnecting analysis of the whole genome sequencing and performed phenanthrene degradation experiments yielded a certain understanding of the bacterial mechanisms involved. The strain isolated in this study was compared to its GC content within its genus. This strain had a GC content of 69.09% against *D. psychralacaliphila* ILA-1 (69.6%), *D. maris* (70.4%), and *D. natronolimnaea* (66.1%) [38,39]. The whole genome was 4,216,480 bp in length containing 3902 CDSs when annotated by the PROKKA annotation tool.

The property of emulsification was assessed in terms of quantifying the E_24_ index. Production of biosurfactants is crucial for bacteria to survive in toxic environments. The emulsifying properties of the bacteria provided by the biosurfactants aid in the solubility of the hydrocarbons, leading to a reduction in the interfacial surface tension, rendering them available for the microbial degradation process [40]. *D. psychralcaliphila* JI1D displayed maximum emulsifying activity with hydrocarbons xylene (63%), benzene (61%), and toluene (59%) when compared to diesel (43%) and cooking oil (41%). The positive control displayed comparable emulsifying properties when compared with the hydrocarbon substrates. When compared with the positive control, samples xylene > benzene > toluene was found to display activity in the range 55–65%. For the cooking oil and diesel, the sample showed lesser activity when compared with their respective positive controls. This property has also been reported for other strains of this genus. With respect to the monoaromatic compound toluene, *D. psychralcaliphila* JI1D showed an E_24_ value of 59% similar to that of *D*. *maris* As-13-3, which showed 60.29%. On the contrary, for diesel, *D. psychralcaliphila* JI1D showed lower activity (27%) than *D. maris* As-13-3 (63.64%) [17]. Moreover, bacteria representing *Dietzia* sp. CN-3, *D*. *maris* As-13-3, and *D cinnamea* KA-1 showed biosurfactant activity when grown under their respective conditions [19,41,42]

The degradation ability of phenanthrene was analyzed by HPLC. *D. psychralcaliphila* JI1D showed 85% degradation of phenanthrene when grown in MSM media. It was interesting to note that the degradation had come to a halt after the 7th day and the degradation rate (R_x_) was 13% of the average degradation efficiency (84.66%). This could be due to the accumulation of by-products generated during the course of the degradation [43], leading to lesser accessibility to the substrate for metabolism. Fathi et al. reported 73% degradation of phenanthrene mediated by the *D. cinnamea* AP strain at the end of 10 days equivalent to 2.0 mM of phenanthrene concentration [44]. It is evident that *D. psychralcaliphila* JI1D is efficient to thrive in a higher concentration of phenanthrene. Degradation of phenanthrene occurs through the C 1,2, C 3,4, and C 9,10 positions of the molecule phenanthrene mediated by oxygenases, generating dihydroxy phenanthrenes and hydroxy phenanthrenes [10]. The presence of numerous oxygenase, including monooxygenase and dioxygenase, in the genome highlights that *D. psychralcaliphila* JI1D is a versatile degrader of PAHs [45]. Notably, the Rieske 2Fe-2S domain containing protein involved in oxygenation reaction, p-cumate 2,3-dioxygenase system, and large component (DOCFNAII_01294) resembled *Rhodoccoccus erythropolis* (94.76%); whereas putative 3-ketosteroid-9-alpha-monooxygenase, oxygenase component (DOCFNAII_02367) resembled *Janitobacter indicus* (78.57%), indicating that the genome has the efficiency in acquiring unique genes to adapt to dynamic environments.

The analysis of generated peaks by GC-MS showed the intermediate products, phthalic acid (J) and salicylic acid (U) (Figure 9), involved in the central pathway of phenanthrene degradation. Other metabolites involving in the degradation process were not detected which could be due to their lower presence or unstable intermediates (which cannot be detected by GC-MS). Phthalic acid and salicylic acid were identified as metabolic intermediates, also in earlier studies [46,47]. The detection of metabolites was analyzed for availability in the whole genome sequence. The probable genes involved in the transformation of the immediate substrate towards the generation of the products (phthalic acid and salicylic acid) have only been discussed. The presence of catabolic gene aldehyde-dehydrogenase *phd*K-like (DOCFNAII_01122, DOCFNAII_01146), consisting of the cd07107 conserved domain, which is NADP+ dependent, converts the aromatic aldehydes into the respective carboxylic acids [48]. In this context, 2-carboxybenzaldehyde (I) was converted to give o-phthalic acid (J), probably mediated by *phd*K-like genes (Figure 9). In addition, the catabolic gene belonging to 2-hydroxychromene-2-carboxylate isomerase/*Dsb*A-like thioredoxin domain (DOCFNAII_00917) has the cd00952 conserved domain, which is involved in converting trans-o-hydroxybenzylidenepyruvate (S), leading to salicyaldehyde (T) and pyruvate (Figure 9) [48]. The presence of succinate-semialdehyde dehydrogenase (DOCFNAII_02167) contains the cd07105 conserved domain (*Dox*F) and could be involved in transforming salicyladehyde (T) into salicylic acid (U) (Figure 9). The two reactions mark the degradation through the naphthalene pathway. The functional annotation of the genome involving COG was performed and the categories related to PAH degradation were identified and presented below. COG analysis showed the presence of COG2146, indicating the gene *Nir* D (DOCFNAII_00199 and DOCFNAII_02996) consisted of 2Fe-2S domains [16]. The COG4638 presence (DOCFNAII_02367, DOCFNAII_02907) shows the ability of an initial oxygenation reaction of the PAH degradation. COG0654 confirmed the presence of the *nah*G gene represented by salicylate hydroxylase (DOCFNAII02482). The central pathway of the phenanthrene degradation branches out to the lower pathway, which is basically catechol-based derivatives. The lower pathway consisted of the genetic machinery for the benzoate degradation via the hydroxylation process. Particularly, the cluster genes *benABCD* and *catABC* were present, as shown in Appendix A. Thus, the whole genome analysis for phenanthrene degradation showed the presence of a catabolic funnel involving the initial, central, and lower pathways through which the initial substrate has been broken into its respective by-products, as shown in Figure 9.

The genome when analyzed for viewing the genomic islands showed the possession of more than 14% of the total genes in genomic islands. Several gene clusters were found in the GI view for PAH degradation. The studied strain contained 3961 genes more than earlier reported in *D. psychralcaliphila* ILA1 [36], explaining the fact that the bacterial genome is provided with genomic modules that may not be available in other strains of the same species [49]. Horizontal gene transfer proved to be a common phenomenon exhibited by microorganisms arising from extreme environments through which they acquire genetic information from closely or phylogenetically distinct species for their survival [50].

There are several families of transcriptional regulators involved in aromatic degradation. Aromatic catabolic degradation is not regulated by a single family of regulators but rather by many different families of regulators, which are not specific to the biodegradation process. The genome analyzed for the transcriptional regulators of COGs is under the class K (Appendix A.). The regulators inducing transcription initiation of the catabolic genes that interact with the aromatic substrate were identified. It is important to note that the regulation of the catabolic pathway is very helpful in unravelling the molecular patterns in the process, signals triggering the pathway expressions, and precisely highlights the activations/repression mechanisms [51]. Nine proteins coding for *Lys*R-type transcriptional regulators (LTTRs) (COG0583) indicates their role in regulation of aromatic compounds metabolism and quorum sensing [52]. Six proteins encoding IclR family transcriptional regulators (COG1414), which are described to be similar in structure to LTTRs, often act as catabolic pathway activators for the aromatic compound degradation [53]. Twelve proteins belonging to the multiple antibiotic resistance regulator MarR family (COG1846), and two proteins belonging to the GnTR transcriptional regulators (COG1802) were found [15]. Besides the aforementioned regulatory patterns, the bacteria also contain enormous gene regulators for response regulations, stress regulators, and transportation regulators across cell membranes, indicating that it has a strong capability to accommodate itself in extremely toxic/harsh environments.

## 5. Conclusions

Antarctic marine environments were found to be a source of novel microbes, possessing greater catabolic activity and which are dynamic in nature. PAHs are recalcitrant organic pollutants in the environment that can cause adverse effects on human and animals. The isolation of bacteria from Deception Island was performed regarding their ability to grow on phenanthrene. Through whole genome sequencing, the isolated marine bacterium *D. psychralcaliphila* JI1D demonstrated its capabilities to grow and mineralize hydrocarbon compounds. The assessment towards degradation of phenanthrene based on genetic and metabolite profiles suggests involvement of multiple degradative pathways. This versatile property of the genome makes this strain a suitable candidate for bioremediation processes.

## Figures and Tables

**Figure 1 microorganisms-09-01327-f001:**
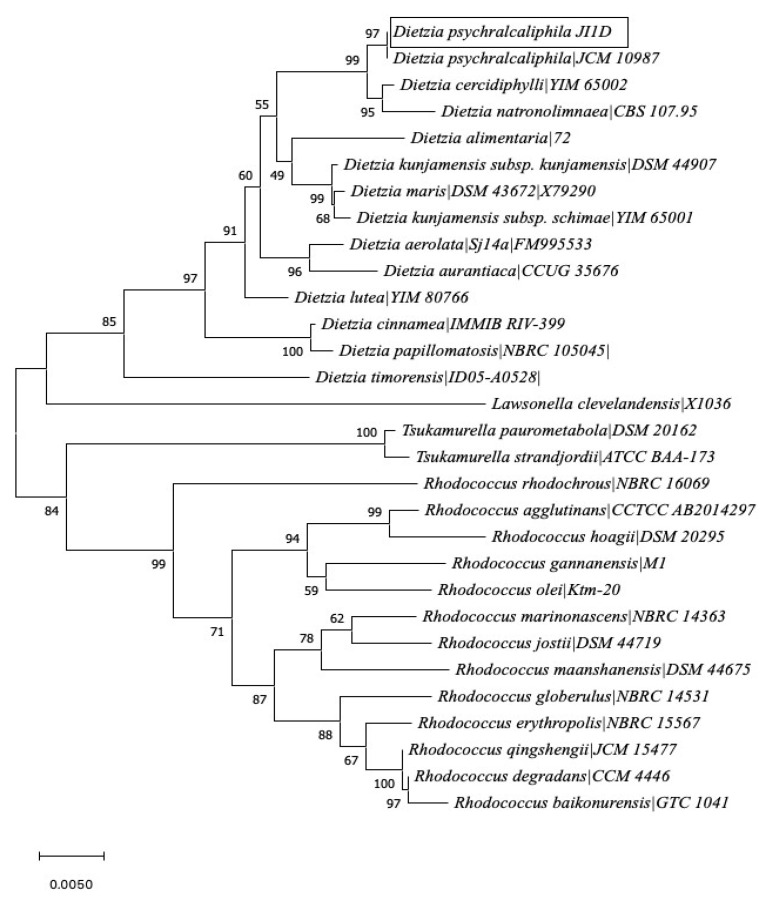
Evolutionary relationship of *D. psychralcaliphila* JI1D predicted by 16S rRNA sequencing.

**Figure 2 microorganisms-09-01327-f002:**
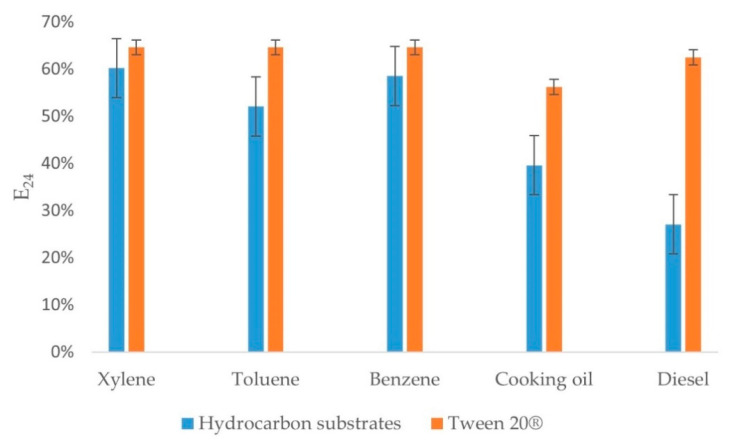
E_24_ assay results on different hydrocarbon substrates, such as xylene, toluene, benzene, cooking oil, and diesel. Tween 20^®^ (Sigma-Aldrich, Darmstadt, Germany) is used as the positive control. Error bars represent the standard deviation values of three independent measurements.

**Figure 3 microorganisms-09-01327-f003:**
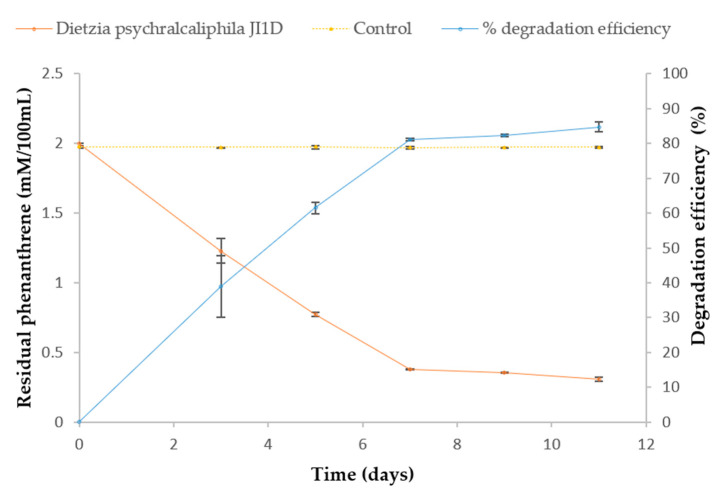
Amount of phenanthrene metabolized by *D. psychralcaliphila* JI1D measured in HPLC over a period of 11 days. Error bars represent the standard deviation of three independent measurements.

**Figure 4 microorganisms-09-01327-f004:**
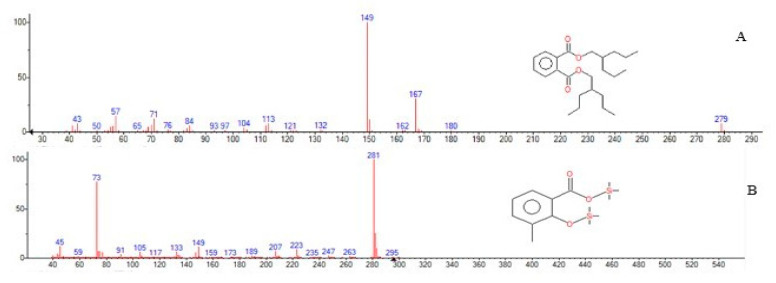
Depicting the GC-MS intermediates of phenanthrene degradation. The indicated fragmentation pattern shows (**A**) phthalic acid, di(2-propylpentyl) ester; (**B**) 3-methylsalicylic acid, 2TMS derivative.

**Figure 5 microorganisms-09-01327-f005:**
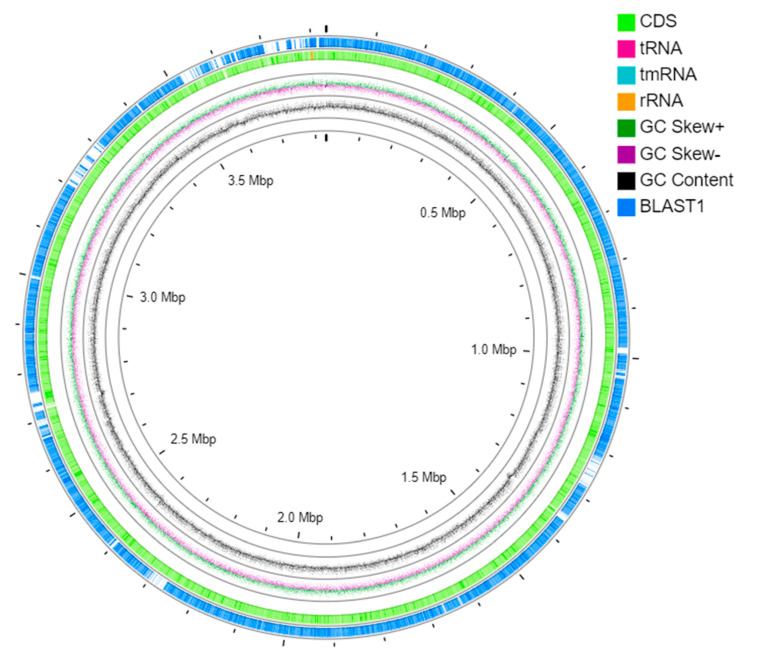
The circular view of the genome *D. psychralcaliphila* JI1D using CGViewer.

**Figure 6 microorganisms-09-01327-f006:**
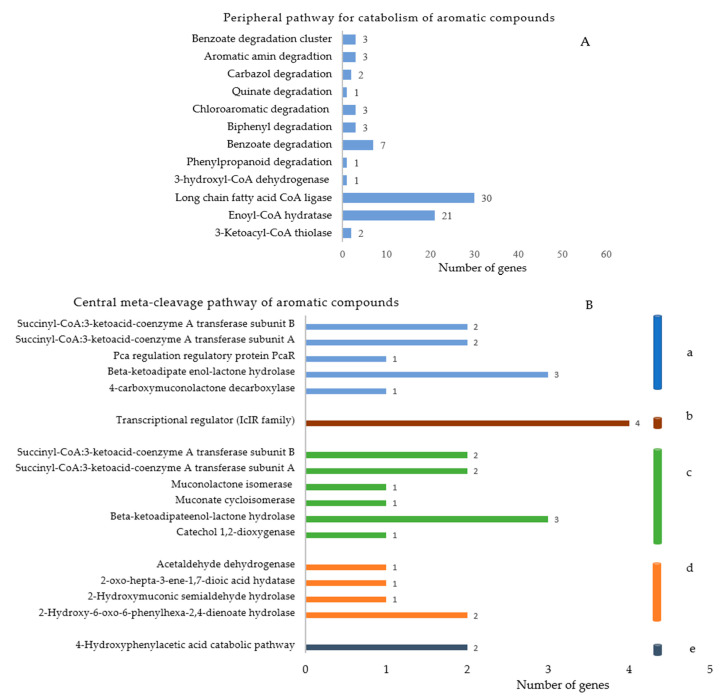
Number of genes involved in the catabolism of the aromatic compounds in *D. psychralcaliphila* JI1D by the peripheral (**A**) and central meta-cleavage (**B**) pathways, respectively (Table 2). Hydroxyphenyl acetic acid catabolic pathway.

**Figure 7 microorganisms-09-01327-f007:**
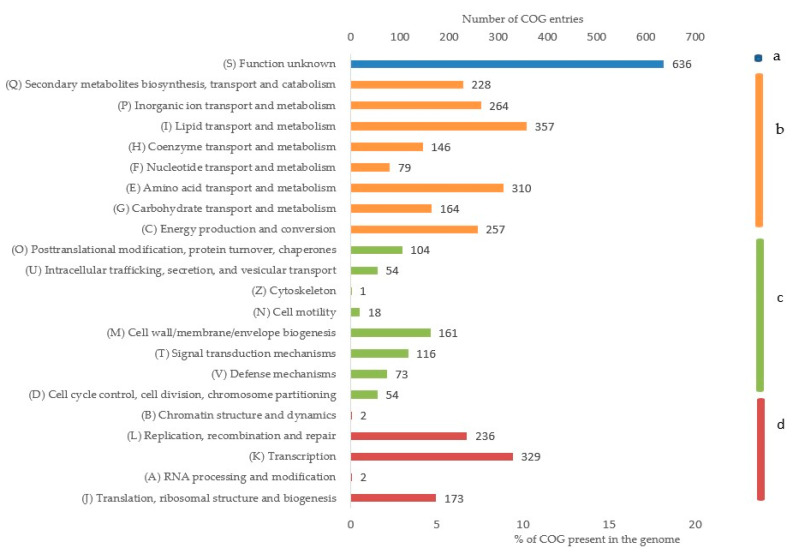
The cluster of ortholog groups (COGs) present in *D. psychralcaliphila* JI1D involved in unknown functions (**a**), metabolism (**b**), cellular and signaling processes (**c**), and genetic information processes (**d**).

**Figure 8 microorganisms-09-01327-f008:**
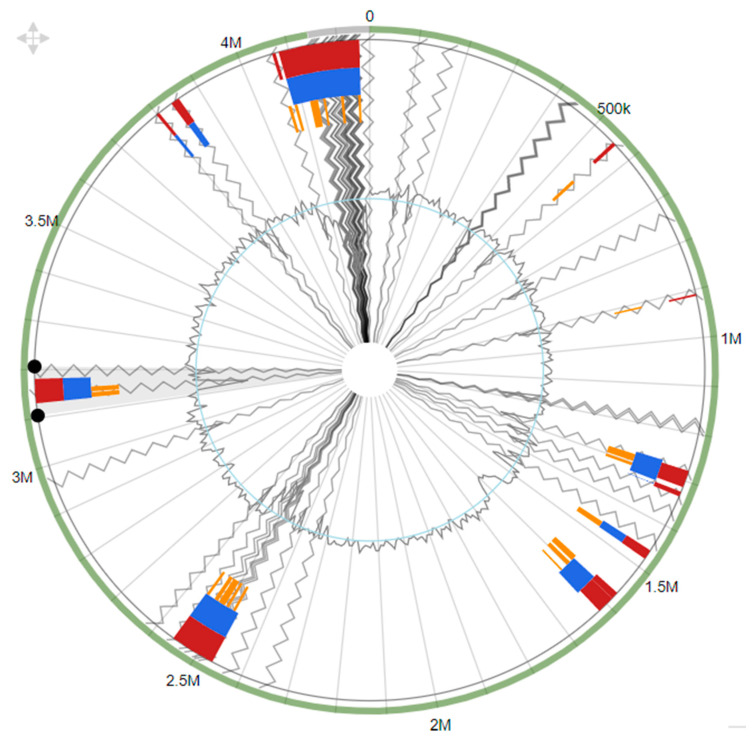
Genomic islands predicted by Integrated (red), IslandPath-DIMOB (blue), SIGI-HMM (orange), and contig boundary (grey lines).

**Figure 9 microorganisms-09-01327-f009:**
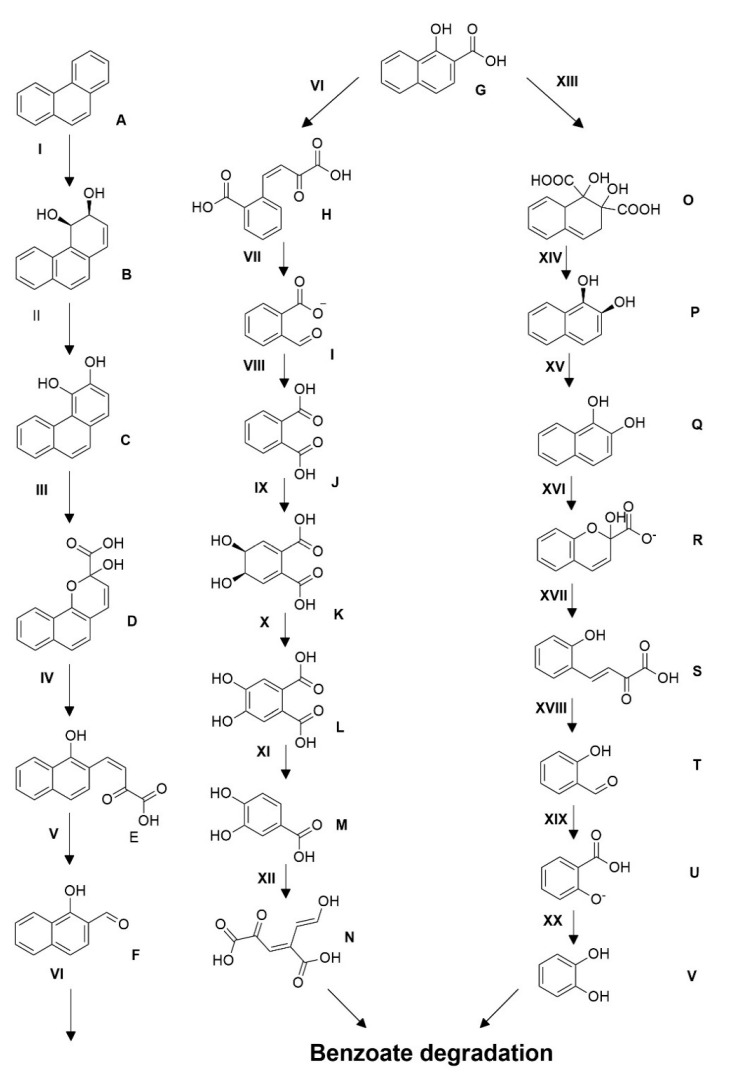
Proposed pathway for phenanthrene degradation by *D. psychralcaliphila* JI1D. The legend for the figure is provided in Appendix A.

**Table 1 microorganisms-09-01327-t001:** Whole genome statistics.

Feature	Statistics
Total genome size including gaps (bp)	4,216,480
Number of contigs	52
Ratio of bases that have phred quality score of over 20 (%)	96.08
Ratio of bases that have phred quality score of over 30 (%)	88.96
Contig N50 (bp)	270,109
Average contig length (bp)	81,086
Longest contig length (bp)	601,190
Shortest contig length (bp)	1026
GC (%)	69.09

**Table 2 microorganisms-09-01327-t002:** Contents of the annotated genome.

Content	Statistics
Total number of genes predicted	3961
Number of genes assigned with CDSs	3902
Number of genes assigned to UniprotKB	1899
Number of genomic islands assigned	29
tRNA	55
rRNA	4 (2 × 5S rRNA,1 × 23S rRNA,1 × 16S rRNA)
ncRNAs	21
Number of genes with non-hypothetical function	2514
Number of genes with EC-number	1611
Number of genes with Seed Subsystem Ontology	1245
Average protein length (aa)	325

## Data Availability

This Whole Genome Shotgun project has been deposited at DDBJ/ENA/GenBank under the accession JAEKIZ000000000. The version described in this paper is version JAEKIZ010000000.

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
