# Peer review of "Assessment of the Degradation Potential and Genomic Insights towards Phenanthrene by Dietzia psychralcaliphila JI1D"

_microorganisms, 2021, doi:10.3390/microorganisms9061327_

Round 1
Reviewer 1 Report
Generally speaking, this is an interesting and innovative paper, which helps to unravel the main degradation pathways of phenantrene. The powerful bioinformatic tools used have allowed the authors to go in depth on the genetic pool associated to hydrocarbons metabolization, but some times the completeness of data is excessive., section 3.7 being the most glaring example. Maybe it will be wise to reduce such contents, as well as the deletion of Table 1, since this information is provided in the text.
Minor corrections on wording:
- 414: “Previous studies have been confirmed that…”
- 484-486: “When the genoma analyzed for this pathway, the genetic machinery for the benzoate degradation via hydroxylation was presente”
- 499-500: “Aromatic catabolic degradation is not regulated by single family of regulators rather by many different familia of regulators…
Author Response
Detailed Response to Reviewers
Note to Editor: We have addressed all of the comments of Reviewers. Our responses are provided in blue font and explain associated changes made to the manuscript.
REVIEWER 1
Generally speaking, this is an interesting and innovative paper, which helps to unravel the main degradation pathways of phenantrene. The powerful bioinformatic tools used have allowed the authors to go in depth on the genetic pool associated to hydrocarbons metabolization, but sometimes the completeness of data is excessive., section 3.7 being the most glaring example. Maybe it will be wise to reduce such contents, as well as the deletion of Table 1, since this information is provided in the text.
We thank the Reviewer for pointing this out. We agree that information in Table 1 is provided in the text, therefore we deleted it in the new version. On the contrary, we think that Section 3.7 is very important to establish the genomic insights for the genes involved in degradation process. In any case, we deleted certain excessive results that were mentioned (lines 453-455 and 469-472, new version).
Minor corrections on wording:
- 414: “Previous studies have been confirmed that…”
Authors are grateful to the reviewer for this suggestion. We changed to “The earlier studies confirmed….” (line 489, new version).
- 484-486: “When the genome analyzed for this pathway, the genetic machinery for the benzoate degradation via hydroxylation was present”
We thank the Reviewer for pointing this out. We changed to “The lower pathway consisted of the genetic machinery for the benzoate degradation via hydroxylation process” (lines 567-569, new version).
- 499-500: “Aromatic catabolic degradation is not regulated by single family of regulators rather by many different families of regulators…
We agree, the sentence was changed to “There are several families of transcriptional regulators involved in aromatic degradation” (lines 583-584, new version)
REVIEWER 2
This manuscript describes a study on phenanthrene degrading strain isolated from marine sediments collected in Antarctica. The authors described the isolation process, molecular identification, degradation analysis, emulsification ability, GC-MS study before reporting on the findings from WGS study. Overall, there were more than enough findings to report for this manuscript, however, the presentation of the studies was not organized and it seems that the authors lose focus on the main objectives several times throughout the manuscript. The problems are as following:
- What were the objectives for this manuscript? I believe there was a lot of emphasis on phenanthrene degradation by strain JI1D. If that was the case, why is the novelty of this strain relevant in this manuscript? What is the significance of figure 1 and 2 in regards to phenanthrene degradation? Furthermore, strain JI1D cannot be declared as a novel strain based on genomic data alone.
We thank the Reviewer for pointing this out. We agree that the objective of this manuscript is to demonstrate the strain JI1D is a potential candidate for bioremediation of PAHs and, therefore, to emphasize the PAHs degradation. To prove its candidacy, we here attempted a multidisciplinary approach involving the analytical results and that of whole genome data. Dietzia is a lesser studied/no data available with its genomic analysis with respect to phenanthrene degradation. On the contrary, we disagree that the novelty of the strain is not relevant, as well as the taxonomic relationship with other genus of bacteria (Figure 1). We think that a greater knowledge of the strain is essential before studying its characteristics.
In any case, considering the referee's thought, we reduced the information about this aspect in the main text, moving Figure 2 to the supplementary.
Moreover, we agree that the strain cannot be declared as a novel strain based on genomic date alone and, therefore, we are grateful to the reviewer for this comment. In fact, the novelty was investigated by FastANI tool, and it is evident from earlier studies that values >95% “might” represent different strain (Jain et al., 2018. DOI: 10.1038/s41467-018-07641-9). For this reason, we modified the text of the revised version adding “might” to the sentences: “the selected bacterium might belong to a novel strain…” (line 84, new version); “D. psychralcaliphila JI1D in this study might represent a novel strain” (lines 244-245, new version).
- Authors claimed the stain isolation was conducted based on the ability to use phenanthrene as sole carbon source. However, yeast extract was included in the MSM media used for isolation. Does this provide an alternative carbon source for the strain?
We thank the Reviewer for pointing about basis for addition of yeast extract. The yeast extract used is from Conda pronadisa, CAT- 1702.00. This acts as the nitrogen source for the media. The contents of this yeast extract are total nitrogen, aminic nitrogen, chlorides and moisture (https://www.condalab.com/int/en/peptones-and-extracts/1256-12431-yeast-extract.html), therefore we can consider phenanthrene as the only carbon source. Gallego et al., 2014 (DOI 10.1007/s10532-013-9680-z), is just an example work where yeast extract was used as nitrogen source in respect with phenanthrene as sole carbon source.
- E24 assay is for measuring emulsification ability, and it is best presented when compared with a control strain (positive control) for comparison. This was also not discussed further in relation to other published studies.
We strongly agree with the Reviewer. The data have been provided now, in methods (lines 140-141), results (264-265), and discussion (line 510-514, 515-518) of the revised version. Figure 3 was also modified.
- For the WGS methodology (section 2.71), many details were not written (coverage of reads, genome assembly approach). Did the de novo assembly yielded a complete genome? In table 3. showed 52 contigs, hence incomplete genome?
We thank the Reviewer for pointing this out. The de novo assembly yielded a draft genome. The sequence reads were assembled at a contig level. Overall, the genome representation is full. The validation method is mentioned in line 194. The details have been provided in the Supplementary, new Table S3. A new paragraph was also added in the Section 3.4 (lines 317-319, new version)
- There were many genes discovered related to aromatic compounds degradation, however, discovery of gene sequences is not sufficient to confirm involvement in degradation pathways.
Authors are grateful to the reviewer for pointing this out. The genes discovered are clear indication of the capability of the strain to undergo aromatic compound degradation. The results are also supported with presence of conserved domains (cd) (found in those gene sequences) and Cluster of Ortholog Groups (COG) analysis. Anyway, we agree with the reviewer that it is not sufficient to confirm the involvement in degradation pathway. Therefore, we rewrite the Section 3.7 (lines 448-466, 546, 552 and 557) and the discussion in new version describing a “probable” pathway.
Now, we do hope that the manuscript is suitable for publication on Microorganisms
Yours sincerely

Reviewer 2 Report
This manuscript describes a study on phenanthrene degrading strain isolated from marine sediments collected in Antarctica. The authors described the isolation process, molecular identification, degradation analysis, emulsification ability, GC-MS study before reporting on the findings from WGS study. Overall, there were more than enough findings to report for this manuscript, however, the presentation of the studies were not organized and it seems that the authors lose focus on the main objectives several times through out the manuscript. The problems are as following:
- What were the objectives for this manuscript? I believe there was a lot of emphasis on phenanthrene degradation by strain JI1D. If that was the case, why is the novelty of this strain relevant in this manuscript? What is the significance of figure 1 and 2 in regards to phenanthrene degradation? Furthermore, strain JI1D cannot be declared as a novel strain based on genomic data alone.
- Authors claimed the stain isolation was conducted based on the ability to use phenanthrene as sole carbon source. However, yeast extract was included in the MSM media used for isolation. Does this provide an alternative carbon source for the strain?
- E24 assay is for measuring emulsification ability and it is best presented when compared with a control strain (positive control) for comparison. This was also not discussed further in relation to other published studies.
- For the WGS methodology (section 2.71), many details were not written (coverage of reads, genome assembly approach). Did the de novo assembly yielded a complete genome? In table 3. showed 52 contigs, hence incomplete genome?
- There were many genes discovered related to aromatic compounds degradation, however, discovery of gene sequences are not sufficient to confirm involvement in degradation pathways.
A thorough rewrite (reorganization of contents) is necessary.
Author Response

(The authors gave the same response as above.)

Round 2
Reviewer 2 Report
Dear authors,
I just have a couple of minor corrections, as below:
Line 122: For MgCl2, 2 should be written in subscript
Line 123: For PolyTaq DNA polymerase, please include manufacturer details and country
As for the yeast extract, I visited the URL provided and from the manufacturer's data sheet, you can see there are proteins (more than 60%) listed in the description of the yeast extract. And there are a lot more carbon in proteins than nitrogen. I understand that yeast extract powder are generally used as nitrogen source in microbiology and usually there is no issue. However, when you stated phenanthrene as "sole carbon source", then I have to disagree. It would be only technically correct if you were to use ammonium salt as nitrogen source rather than yeast extract to claim as "sole carbon source". You can try a simple experiment by omitting phenanthrene from your MSM medium (which includes yeast extract), and check whether strain JI1D would still grow. I am very confident it will grow quite well. I do not intend to deny the strain's ability to utilize phenanthrene, but I believe "sole carbon source" is not accurate for this case.
Author Response
Response to Reviewer
Note to Editor: We have addressed all of the comments of Reviewers. Our responses are provided in blue font and explain associated changes made to the manuscript.
REVIEWER 2
Dear authors,
I just have a couple of minor corrections, as below:
Line 122: For MgCl2, 2 should be written in subscript
We thank the Reviewer for pointing this out. Mgcl2 have been replaced with MgCl2 in the new version of the manuscript (line 121).
Line 123: For PolyTaq DNA polymerase, please include manufacturer details and country
Authors are grateful to the reviewer for this suggestion. The details of the manufacturer for the Poly Taq DNA polymerse have been added (lines 122-123, new version).
As for the yeast extract, I visited the URL provided and from the manufacturer's data sheet, you can see there are proteins (more than 60%) listed in the description of the yeast extract. And there are a lot more carbon in proteins than nitrogen. I understand that yeast extract powder are generally used as nitrogen source in microbiology and usually there is no issue. However, when you stated phenanthrene as "sole carbon source", then I have to disagree. It would be only technically correct if you were to use ammonium salt as nitrogen source rather than yeast extract to claim as "sole carbon source". You can try a simple experiment by omitting phenanthrene from your MSM medium (which includes yeast extract), and check whether strain JI1D would still grow. I am very confident it will grow quite well. I do not intend to deny the strain's ability to utilize phenanthrene, but I believe "sole carbon source" is not accurate for this case.
We thank the Reviewer for this comment regarding the involvement of yeast extract. The authors agree with reviewer point of view. Therefore, the word “sole carbon source” has been removed for the manuscript in the new version (lines 47,109-110, 216-217, 343-344, 418, 535-536).
